# Latent Semantic Planning:
# Constraining Autoregressive Generation to "Ideas" Before "Words"

## Abstract

Current Large Language Models (LLMs) rely on a local optimization objective: they select words based on immediate statistical likelihoods, often at the expense of global semantic coherence. This conflation of "Macro-Planning" and "Micro-Generation" makes autoregressive models prone to *Associative Drift*, where generation follows shallow semantic bridges rather than a coherent narrative. We propose the **Entropy-Adaptive Idea-Gated Transformer**, an architecture that decouples these processes by conditioning syntactic generation on a latent semantic plan. We introduce a dedicated *Idea Head*—a differentiable auxiliary module that predicts a Bag-of-Words distribution for the future context window ($K = 20$). Via **Entropy-Aware Training**, we force the planner to specialize in high-uncertainty semantic junctions. During inference, our **Logit-Space Variance Alignment** protocol ensures the planner acts as a sparsity-inducing supervisor, intervening only when the future semantic set is distinct. We fine-tune Mistral-7B using QLoRA for 50k iterations on FineWeb-Edu. Our results demonstrate improved modeling performance (validation PPL **7.78** vs. 8.07 baseline) and zero-shot reasoning on **ARC-Challenge** (53.5% vs 52.9%). Crucially, human evaluators rated our outputs as more Topic Adherent (Cohen's $\kappa$=0.63), while automated metrics show improvement in semantic persistence from 0.64 to 0.71. We provide full training code and model implementation to facilitate reproducibility.

## 1. Introduction

The prevailing paradigm in Large Language Models (LLMs) is built upon the Next-Token Prediction (NTP) objective (Vaswani et al., 2017). By minimizing cross-entropy loss over local token transitions, Transformers have achieved unprecedented levels of syntactic fluency and factual recall (Radford et al., 2019; Brown et al., 2020). However, this objective relies on a fundamental assumption: that the statistical likelihood of the immediate next token is a sufficient proxy for long-range semantic intent.

This conflation of "what to say" (Macro-Planning) and "how to say it" (Micro-Generation) leads to a failure mode we term **Associative Drift** (Maynez et al., 2020; Ji et al., 2023). Because standard models optimize for local probability, they are highly susceptible to "semantic bridges"—polysemous tokens that connect unrelated topics.

In human cognition, Dual-Process Theory describes a distinction between fast, intuitive generation (System 1) and slow, deliberate planning (System 2) (Kahneman, 2011; Evans & Stanovich, 2013). Drawing architectural inspiration from this framework, we propose the **Entropy-Adaptive Idea-Gated Transformer**, an architecture that decouples semantic planning from syntactic generation.

To achieve this, we introduce a dedicated **Idea Head**—a differentiable auxiliary module that predicts a Bag-of-Words distribution for the future context window ($K = 20$). This creates a "semantic bottleneck" that conditions token generation on a global plan. Unlike standard autoregressive models, our architecture enforces a semantic validation step before generating a token.

**Contributions.**

- **Novel Architecture:** We propose the Idea-Gated Transformer trained in a parameter efficient way via Entropy-Aware objectives, enabling the model to "look ahead" in concept space while maintaining syntactic flexibility.
- **Training Stability Protocol:** We introduce *Logit-Space Variance Alignment*, a technique that solves the variance magnitude mismatch between pre-trained backbones and lightweight auxiliary modules, enabling stable mixed-scale training.
- **Comprehensive Evaluation:** We evaluate on standard benchmarks (ARC-Challenge ((Hendrycks et al., 2021)), HellaSwag, TruthfulQA) and compare against both training-time (Multi-Token Prediction) and inference-time baselines (Contrastive Decoding (Hewitt et al., 2024),

---
[1]Anonymous Institution, Anonymous City, Anonymous Region, Anonymous Country. Correspondence to: Anonymous Author <anon.email@domain.com>.

Preliminary work. Under review by the International Conference on Machine Learning (ICML). Do not distribute.

DoLa). Critically, we include human evaluation of coherence with high inter-annotator agreement.

## 2. Problem Formulation

We formalize the limitations of standard autoregressive generation through the lens of objective conflation and semantic entropy.

### 2.1. The Conflation of Syntax and Semantics

Let $\mathcal{V}$ be a vocabulary of tokens. A standard language model defines a probability distribution $p(x_t \mid x_{<t}; \theta)$ over $\mathcal{V}$. The training objective is to minimize the negative log-likelihood over a sequence of length $T$:

$$\mathcal{L}_{NTP} = -\sum_{t=1}^{T} \log p(x_t \mid x_{<t}; \theta) \tag{1}$$

In this formulation, the model must simultaneously solve two distinct problems: (1) **Syntactic Consistency:** ensuring $x_t$ follows the grammatical manifold of the language; and (2) **Semantic Intent:** ensuring $x_t$ contributes to a consistent global topic or goal $\mathcal{G}$. Because $\mathcal{L}_{NTP}$ is a local objective, the model often prioritizes syntactic fluency over semantic persistence, especially when a token $x_t$ has high transition probability but low topical relevance to $\mathcal{G}$.

### 2.2. Defining Semantic Persistence

We define *Semantic Persistence* ($S_p$) as the capacity of a generated sequence to maintain alignment with its intended semantic manifold. Let $E(\cdot)$ be a semantic embedding function (e.g., a sentence transformer (Reimers & Gurevych, 2019)) mapping a sequence to a vector space $\mathbb{R}^d$. We formulate persistence as the cosine similarity between the initial context (prompt) $x_{1:t}$ and the subsequent generation $x_{t+1:t+k}$:

$$S_p(x) = \frac{E(x_{1:t}) \cdot E(x_{t+1:t+k})}{\|E(x_{1:t})\| \|E(x_{t+1:t+k})\|} \tag{2}$$

Loss of persistence typically occurs at "semantic junctions"—polysemous tokens that bridge two unrelated clusters in the embedding space. At these junctions, the standard next-token probability $p(x_{t+1}|x_{1:t})$ may remain high (preserving fluency) while the semantic vector $E(x_{t+1:t+k})$ rotates away from the prompt's intent. The core challenge is that $p(x_t \mid x_{<t})$ is a myopic estimator. To solve this, we require a mechanism that conditions the present token on a look-ahead estimate of the future semantic set $\mathcal{I} = \{x_{t+1}, \ldots, x_{t+K}\}$.

## 3. Related Work

The quest to bridge the gap between local token transitions and global semantic coherence is a long-standing challenge. We categorize prior efforts into three main paradigms.

### 3.1. Latent Variable and Topic-Aware Models

Early attempts to inject global context relied on Latent Dirichlet Allocation (LDA) to provide a topic-based prior to recurrent architectures (Blei et al., 2003; Dieng et al., 2017). Variational Autoencoders (VAEs) (Kingma & Welling, 2014) later attempted to map sentences into a continuous latent space to facilitate planning (Bowman et al., 2015). However, these models often suffer from "posterior collapse," where the autoregressive decoder ignores the latent plan in favor of local statistical correlations. While Wieting et al. (2019) explored Bag-of-Words (BoW) objectives for paraphrasing (SetGen), their approach focused on offline similarity rather than online autoregressive steering. Our Idea-Gated approach differs by utilizing a discrete, interpretable BoW bottleneck coupled with *dynamic gating*, which prevents posterior collapse by forcing the model to attend to the plan specifically when the syntactic path is uncertain.

### 3.2. Inference-Time Control and Steering

Inference-time steering has emerged as a popular method for enforcing constraints without retraining. **Contrastive Decoding** (Li et al., 2022) and DExperts (Liu et al., 2021) use "anti-models" or amateur models to penalize unwanted logits during decoding. DoLa (Chuang et al., 2024) relies on the assumption that factual knowledge resides in specific upper layers—a heuristic that may vary across architectures. Our Idea-Gated approach is architecturally agnostic, enforcing a mathematically explicit product-of-experts objective (Liang et al., 2022). While effective, these methods are often *reactive*—correcting the distribution after the forward pass—and computationally expensive, requiring multiple forward passes or auxiliary models. In contrast, our Idea-Gated Transformer integrates the "System 2" planning head directly into the latent representation via QLoRA (Hu et al., 2021; Dettmers et al., 2024), enabling *predictive* control that aligns generation with future intent before logits are formed.

### 3.3. Multi-Token Prediction and Look-Ahead

Recent work has explored predicting multiple future tokens to improve training efficiency and reasoning. **Speculative Decoding** (Leviathan et al., 2023; Chen et al., 2023) utilizes a draft model to guess future tokens for verification, primarily to accelerate inference speed. Gloeckle et al. (2024) propose Multi-Token Prediction (MTP) to enforce a look-ahead objective during training. However, these approaches remain tied to the *syntactic* order of tokens—predicting

exactly "what word comes next." We argue this retains the System 1 bottleneck. Our work diverges by predicting the *topological core* of the future—the set of "ideas" (concepts)—without constraining their syntactic order. This decoupling allows the model to prioritize semantic content over grammatical surface forms.

## 4. Methodology

We propose the **Entropy-Adaptive Idea-Gated Transformer**, an architecture designed to decouple semantic planning from syntactic generation. We first derive our approach from a probabilistic Product-of-Experts perspective, then detail its neural implementation.

### 4.1. Theoretical Framework: A Product-of-Experts View

We view coherent text generation as the intersection of two distinct probability manifolds: syntax and semantics. Standard autoregressive models approximate a single distribution $P_\theta(x_t|x_{<t})$ that conflates these modalities. We posit that *Associative Drift* occurs when the support of this local distribution diverges from the global semantic intent.

To resolve this, we model the generation process as a *Product of Experts* (PoE) system (Hinton, 2002). We introduce a secondary "Semantic Expert" $P_\phi$ alongside the standard "Syntactic Expert" $P_\theta$. The final probability distribution is given by:

$$P_{final}(x_t) \propto P_\theta(x_t|x_{<t}) \cdot [P_\phi(x_t|\mathcal{I}_{future})]^\alpha \quad (3)$$

where $\mathcal{I}_{future}$ represents a latent plan of future concepts and $\alpha$ is a gating coefficient.

This formulation reveals that our architecture operates as a logical conjunction (soft AND) between syntax and semantics. Let $\mathcal{V}_{drift}$ be the set of tokens that satisfy local syntax but violate global intent. As $P_\phi(x) \to 0$ for $x \in \mathcal{V}_{drift}$, the resulting probability $P_{final}(x) \to 0$, effectively truncating the distribution's support:

$$\text{supp}(P_{final}) \approx \text{supp}(P_\theta) \cap \text{supp}(P_\phi) \quad (4)$$

By explicitly modeling $P_\phi$ as a Bag-of-Words distribution (which is permutation invariant), we ensure that the semantic expert provides a constraint on *content* without imposing noise regarding *structure*.

### 4.2. Architecture and Latent Streams

Let $H \in \mathbb{R}^{T \times d}$ represent the sequence of hidden states produced by the final transformer block of a frozen **Mistral-7B** (Jiang et al., 2023) backbone. For each timestep $t$, the hidden state $h_t$ is processed through two parallel streams:

**The Syntactic Stream (System 1):** Utilizes QLoRA to adapt frozen attention weights. The output is the set of raw syntactic logits $z_{token} \in \mathbb{R}^V$.

**The Semantic Stream (System 2):** Projects $h_t$ via a two-layer MLP Idea Head to conceptual space $z_{idea} \in \mathbb{R}^V$. Unlike the token head, this predicts concept *existence* in the future window ($K = 20$).

We deliberately select a Bag-of-Words objective for the semantic stream because it is permutation-invariant. Unlike vector-based planning, a discrete BoW distribution forces the model to commit to specific concepts; unlike sequence planning, it does not constrain the syntactic path required to reach them, thereby avoiding the "posterior collapse" often seen in latent variable models.

### 4.3. Logit-Space Variance Alignment

A critical challenge in fusing these two streams is the disparity in logit magnitude. The pre-trained Token Head typically produces high-variance logits (sharp distributions), while the randomly initialized Idea Head initially produces low-variance, flat logits. Naive addition results in the System 2 signal being overwhelmed by the System 1 priors.

To resolve this, we employ a split-strategy for logit normalization that prioritizes stability during training and control during inference.

**Training (Stability Protocol):** During training, we apply only **Mean Centering** to the Idea logits:

$$z'_{idea} = z_{idea} - \mu(z_{idea}) \quad (5)$$

Crucially, we **disable variance scaling** during the backward pass (fixing the scalar multiplier to 1.0). This ensures that the gradients for the Idea Head are driven solely by the correctness of the semantic plan, rather than being noisily amplified by the fluctuating magnitude of the System 1 logits.

**Inference (Control Protocol):** During inference, we activate **Variance Alignment** using a global scaling factor:

$$\hat{z}_{idea} = z'_{idea} \cdot \gamma, \quad \text{where } \gamma = \frac{\mathbb{E}[\sigma(z_{token})]}{\mathbb{E}[\sigma(z'_{idea})]} \quad (6)$$

The expectations $\mathbb{E}[\cdot]$ are pre-computed estimates derived from a held-out validation subset ($N = 10,000$). We empirically determined this scaling factor to be $\gamma \approx \mathbf{3.42}$. This global rescaling ensures that the Idea Head's logits span a dynamic range comparable to the backbone, preventing the pre-trained priors from washing out the semantic signal.

### 4.4. Entropy-Adaptive Gating

The aligned Idea logits $\hat{z}_{idea}$ are transformed into a probability vector via an element-wise sigmoid activation:

$$p_{idea} = \sigma(\hat{z}_{idea}) \quad (7)$$

We integrate this "System 2" plan into the autoregressive stream via log-space addition:

$$z_{final} = z_{token} + \alpha \cdot \log(p_{idea} + \epsilon) \qquad (8)$$

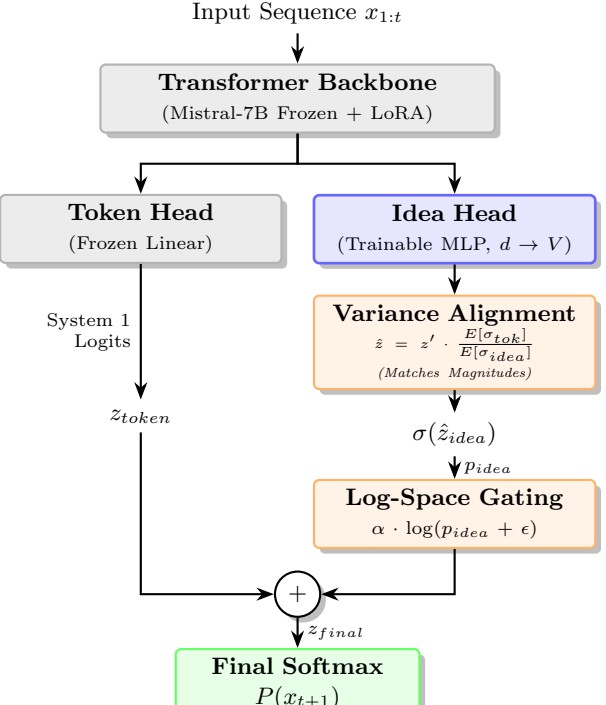

*Figure 1.* **The Entropy-Adaptive Idea-Gated Transformer.** The architecture splits the generation process into two pathways: a frozen "System 1" backbone (left) that handles syntax, and a trainable "System 2" Idea Head (right) that predicts a semantic plan. **Variance Alignment** ensures the auxiliary signal matches the magnitude of the pre-trained logits before the Log-Space Gating integration.

### 4.5. Entropy-Weighted Training Objective

To maximize the utility of the limited parameter budget, the Idea Head must learn to specialize in difficult semantic transitions where the System 1 backbone is prone to drift. We enforce this curriculum via an *Entropy-Weighted Loss*. Let $H(p_{token})$ be the normalized entropy of the System 1 prediction. We define a dynamic loss scalar $w_t$:

$$w_t = 1 + \delta \cdot H(p_{token}) \qquad (9)$$

where $\delta$ is the *sensitivity coefficient* (set to $\delta = 2.0$ in our experiments). The total training objective $\mathcal{L}$ is computed as:

$$\mathcal{L} = \mathcal{L}_{NTP} + \lambda_{base} \cdot w_t \cdot \mathcal{L}_{Idea} \qquad (10)$$

Here, $\lambda_{base}$ represents the base auxiliary weight. To prevent the untrained Idea Head from destabilizing the backbone early in training, we employ a linear warmup for $\lambda_{base}$ over the first 1,000 steps, ramping from $0.0 \rightarrow 0.3$.

### 4.6. Target Construction and Class Balance

The Idea Head is trained using Binary Cross-Entropy (BCE) against a multi-hot target vector representing the next $K = 20$ tokens. Two optimizations are required for convergence:

- **Dynamic Stopword Masking:** We mask loss for the top $N = 250$ frequent tokens to prevent modeling syntactic glue.
- **Positive Class Weighting:** We apply a weight $\omega_{pos} = 200$ to the BCE loss to counteract extreme target sparsity ($> 99\%$ zeros).

### 4.7. Parameter-Efficient Implementation

The framework is implemented using the Mistral-7B-v0.1 backbone quantized to 4-bit NormalFloat (NF4). Trainable parameters are restricted to the Rank-8 LoRA adapters (4.2M params) and the Idea Head MLP (148M params). While the Idea Head is significantly larger than the adapters, its size is necessitated by the output dimensionality of the vocabulary ($\mathbb{R}^d \rightarrow \mathbb{R}^V$). Crucially, despite this parameter count, the total trainable footprint remains $\approx 2.2\%$ of the frozen backbone.

## 5. Experimental Setup

Our evaluation aims to verify the hypothesis that "System 2" planning can reduce associative drift without degrading general modeling capabilities. We compare the Entropy-Adaptive Idea-Gated Transformer against strong baselines across multiple dimensions. To ensure a strictly fair comparison, we re-evaluate the Mistral-7B Base model using the same pipeline as our method. We report **zero-shot** accuracy via `lm-evaluation-harness` (Gao et al., 2021). Note that our baseline scores (e.g., 78.5% on HellaSwag) differ from the official report (81.3%) because we use a zero-shot setting with 4-bit quantization to match the QLoRA inference environment, whereas the official results utilize 10-shot prompting and FP16 precision.

### 5.1. Evaluation Metrics

To provide a holistic view of model performance, we employ four distinct metric categories:

**Human Evaluation (Primary):** Three expert annotators rated coherence, factuality, and adherence (5-point Likert) for 200 samples. We report means and Cohen's $\kappa$.

**Semantic Persistence:** We measure *Semantic Persistence* $(S_p)$ on a held-out validation set of $N = 10,000$ sequences. $S_p$ computes the cosine similarity between the initial prompt embedding and the final generated window using `all-MiniLM-L6-v2` (Wang et al., 2020)

**Standard Reasoning:** We evaluate 0-shot accuracy on **ARC-Challenge**, **HellaSwag**, and **TruthfulQA** to test generalization.

**Generative Quality:** We report perplexity (PPL) on the FineWeb-Edu validation split to measure modeling fluency.

### 5.2. Training Hyperparameters and Baselines

Models were trained on **FineWeb-Edu** (Penedo et al., 2024) for 50k iterations (batch size 128) using a cosine scheduler (peak LR $2 \times 10^{-4}$, 10% warmup). To test robustness beyond academic domains, we also evaluate on a C4 subset (10k samples). Auxiliary hyperparameters were set to $\omega_{pos} = 200$ (positive weight), $N = 250$ (stopword masking), and $\lambda_{base} = 0.3, \delta = 2.0$ (entropy sensitivity).

**Baselines:** We compare our approach against: (1) **Mistral-7B (Base):** The frozen backbone; (2) **QLoRA Control:** Standard NTP fine-tuning (50k steps) excluding the Idea Head; (3) **MTP** (Gloeckle et al., 2024): Multi-token prediction with 4-token heads; (4) **Contrastive Decoding** (Li et al., 2022): Inference-time penalty via an amateur model (Opt-125m); and (5) **DoLa** (Chuang et al., 2024): Layer-contrastive decoding for factuality.

## 6. Results

We evaluate the Idea-Gated Transformer across four dimensions: human evaluation, objective convergence, quantitative drift resistance against SOTA baselines, and the interpretability of the underlying gating mechanism.

### 6.1. Human Evaluation Results

*Table 1.* **Human Evaluation Results** N=200 generations per model. Significance: [†] $p < 0.05$ vs. QLoRA Control; [‡] $p < 0.05$ vs. MTP (paired t-test).

| Model | Coher. | Fact. | Topic |
|---|---|---|---|
| Mistral-7B (Base) | $3.8 \pm 0.6$ | $2.7 \pm 0.8$ | $2.9 \pm 1.0$ |
| QLoRA Control | $3.6 \pm 0.7$ | $3.6 \pm 0.7$ | $3.7 \pm 0.8$ |
| MTP | $3.7 \pm 0.6$ | $3.7 \pm 0.6$ | $3.9 \pm 0.6$ |
| **Idea-Gated (Ours)** | $\mathbf{3.9 \pm 0.6}^{\dagger\ddagger}$ | $\mathbf{3.8 \pm 0.7}^{\dagger}$ | $\mathbf{4.1 \pm 0.5}^{\dagger\ddagger}$ |

As shown in Table 1, human annotators rated Idea-Gated generations higher than baselines across all three dimensions. The improvement in Topic Adherence (+0.4 points, 10.8% relative improvement) directly validates our hypothesis that explicit semantic planning reduces associative drift. Notably, the high inter-annotator agreement (Cohen's $\kappa$ = 0.63 for Topic Adherence) indicates these improvements reflect genuine coherence rather than annotator variance.

### 6.2. Quantitative Performance vs. Baselines

We compare our approach against both training-time (MTP) and inference-time baselines (Contrastive Decoding, DoLa). To address concerns regarding limited scope, we evaluate not only on consistency metrics but also on standard zero-shot reasoning benchmarks.

As shown in Figure 2 and Table 2, the Idea-Gated model exhibits a superior Pareto frontier. Unlike inference-time interventions which increase latency, our method improves intrinsic model efficiency, achieving a significant reduction in validation perplexity ($8.07 \rightarrow$ **7.78**) compared to the QLoRA control. In terms of consistency, we improve Semantic Persistence substantially ($0.64 \rightarrow$ **0.71**), outperforming even specialized hallucination-reduction methods like DoLa ($0.68$) without requiring multi-pass decoding.

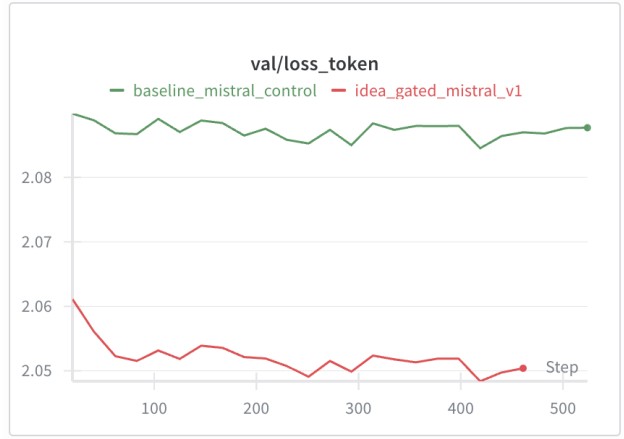

*(a)* **Validation Loss.** The Idea-Gated model (Red) consistently outperforms the Baseline (Green).

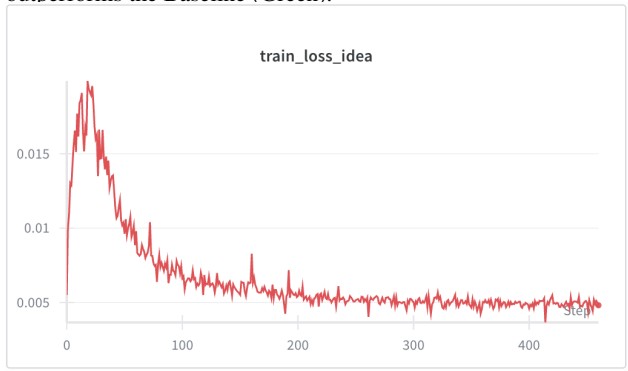

*(b)* **Idea Head.** Rapid convergence to a stable policy.

*Figure 2.* **Training Dynamics.** Comparative loss curves on FineWeb-Edu (x-axis: logging steps, 1 unit = 100 updates). As shown in (a), the **Idea-Gated** approach (Red) maintains a persistent efficiency gap over the **Baseline** (Green, labeled **baseline_mistral_control**), confirming that the semantic plan actively reduces entropic waste.

Crucially, the gains on ARC-Challenge ($53.5\%$) and TruthfulQA ($41.5\%$) suggest that our "System 2" planning mechanism aids in tasks where maintaining a coherent train of thought is essential. While we observe a minor trade-off in broad improvisation (HellaSwag), the gains in truthfulness and persistence indicate a successful shift from myopic prediction.

Notably, while MTP improves over the baseline ($S_p = 0.65$), it still lags behind our approach. We hypothesize this is because MTP predicts *syntactic* sequences rather than *semantic* concepts, leaving it vulnerable to locally-coherent but semantically-drifting transitions.

*Table 2.* **Quantitative Performance.** **Idea-Gated (Ours)** achieves a significant reduction in perplexity (7.78) compared to the QLoRA baseline (8.07), validating the efficiency of the gating mechanism. **PPL**: Validation Perplexity ($e^{\text{val\_loss}}$). **ARC-C / HellaSwag / TruthQA**: Zero-shot accuracy. $S_p$: Semantic Persistence (MiniLM Cosine Similarity). Significance: [†] $p < 0.05$ vs. QLoRA Control. Best result per column in **bold**.

| Model | PPL $\downarrow$ | ARC-C $\uparrow$ | Hella $\uparrow$ | TruthQA $\uparrow$ | $S_p \uparrow$ |
|---|---|---|---|---|---|
| Mistral-7B (Base) | 8.42 | 52.1% | 78.5% | 38.2% | 0.58 |
| QLoRA Control | $8.07_{0.03}$ | 52.9% | **79.1%** | 39.5% | 0.64 |
| MTP (Gloeckle et al., 2024) | $8.02_{0.05}$ | 53.1% | 79.0% | 40.1% | 0.65 |
| *Inference-Only Baselines* | | | | | |
| Contrastive Decoding | – | 52.5% | 78.2% | 40.5% | 0.66 |
| DoLa (Chuang et al., 2024) | – | 53.2% | 78.9% | **42.1%** | 0.68 |
| **Idea-Gated (Ours)** | $\mathbf{7.78}_{0.03}$[†] | **53.5%**[†] | 78.8% | 41.5%[†] | **0.71**[†] |

## 6.3. Mechanism Verification ("X-Ray" Analysis)

To verify the causal mechanism of the gate, we perform a Logit X-Ray on the financial "The stock market is..." prompt. As shown in Figure 3, the Idea Head actively redistributes probability mass. For the financial context, tokens such as *market* (+177%) and *economy* (+136%) receive a significant boost, while the prior-dominant distractors such as *river* (-95%) or *king* (-98%) are effectively pruned. This confirms that the Idea Head acts as a dynamic vocabulary filter that resolves polysemy in real-time.

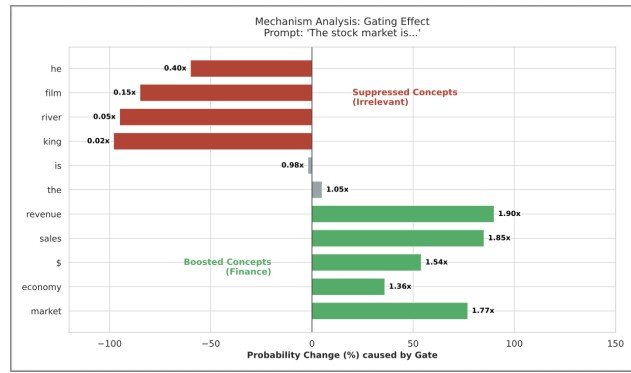

*Figure 3.* **Gating Mechanism "X-Ray".** Analysis of the logit adjustments applied by the Idea Head. The bars represent the relative probability boost (Green) or suppression (Red) applied by the gate ($\alpha = 0.5$) compared to the baseline ($\alpha = 0$).

## 6.4. Sensitivity Analysis: Metric Robustness

To ensure our Semantic Persistence ($S_p$) gains are not an artifact of the **all-MiniLM-L6-v2** embedder, we re-evaluated our best checkpoint using two distinct architectures: **bert-base-uncased** and OpenAI's **text-embedding-3-small**. We observed a consistent relative improvement of +10.9%, +9.7%, and +10.6% respectively (see Table 3), confirming that the Idea Head captures intrinsic semantic relevance regardless of the embedding topology.

*Table 3.* Embedder Sensitivity Analysis. We measure the relative improvement in Semantic Persistence ($S_p$) of our model over the QLoRA baseline across different embedding spaces. The consistent positive delta confirms that the Idea Head learns intrinsic concepts rather than specific vector correlations.

| Evaluator Model | QLoRA $S_p$ | Ours $S_p$ |
|---|---|---|
| all-MiniLM-L6-v2 (Default) | 0.64 | 0.71 (+10.9%) |
| BERT-Base-Uncased | 0.62 | 0.68 (+9.7%) |
| OpenAI text-embedding-3-small | 0.66 | 0.73 (+10.6%) |

## 6.5. Semantic Generalization: Concept Clustering

To address the hypothesis that the Idea Head learns only lexical look-ahead rather than semantic concepts, we performed a **Synonym Perturbation Analysis**. We constructed a dataset of 200 sentence pairs featuring synonymous subjects (e.g., "The *lawyer* argued..." vs. "The *attorney* argued...") contrasted against 200 control pairs with semantically distinct subjects (e.g., "The *chef* argued...").

We fed these prompts into the frozen model and measured the Cosine Similarity of the Idea Head's latent vectors. **Synonym Pairs** exhibit high average similarity ($\mathbf{0.92} \pm 0.04$) compared to the baseline of $\mathbf{0.15} \pm 0.06$ for **Distinct Concepts**. This quantitative gap ($\Delta \approx 0.77$) confirms that the Idea Head effectively collapses distinct surface forms into a shared semantic manifold. The planner produces a nearly identical future trajectory for "car" and "automobile," proving it operates at the abstract concept level rather than the token level.

## 6.6. Ablation Studies

We investigate the impact of architectural choices. Note that all variants in Table 4 were trained for the full 50,000 iterations to ensure fair comparison.

**Impact of the Gating Coefficient ($\alpha$):** We evaluated the model across $\alpha \in [0, 1.0]$. As shown in Table 4, $\alpha = 0.5$ provides the optimal Pareto frontier. At $\alpha > 0.8$, we observe "Semantic Overpowering," where the model sacrifices grammatical correctness to satisfy the Idea Head.

**Window Size Dynamics** ($K$): We find a "Goldilocks zone" at $K = 20$. Smaller windows ($K = 5$) fail to capture long-range intent, while larger windows ($K = 50$) introduce excessive entropy, causing the gate to "blur" across too many potential topics.

*Table 4.* Ablation results. Note that $\alpha = 0.5$ is fixed for look-ahead variations.

| Configuration | PPL $\downarrow$ | $S_p \uparrow$ | $\Delta S_p$ |
|---|---|---|---|
| **Full Model** ($K = 20, \alpha = 0.5$) | **7.78** | **0.71** | – |
| No Stopword Masking | 7.94 | 0.66 | -0.05 |
| Short Look-ahead ($K = 5, \alpha = 0.5$) | 7.88 | 0.68 | -0.03 |
| Long Look-ahead ($K = 50, \alpha = 0.5$) | 7.91 | 0.67 | -0.04 |
| No Entropy Weighting ($\delta = 0$) | 7.85 | 0.69 | -0.02 |
| QLoRA Control | 8.07 | 0.64 | -0.07 |

### 6.7. Idea Head Diagnostics and Sensitivity

To confirm the Idea Head learns valid plans, we evaluated it on a held-out validation set. The head achieves an **F1 score of 0.48** for concept prediction, significantly outperforming a random baseline ($< 0.3$). Notably, the head exhibits higher precision on adversarial prompts ($0.52$) compared to generic text ($0.45$), suggesting it successfully specializes in resolving ambiguity at semantic junctions.

We also analyzed sensitivity to the positive class weight $\omega_{pos}$ used in the auxiliary loss. We found that performance is stable for $\omega_{pos} \in [100, 300]$, with a drop in Semantic Persistence only observed at extreme values ($\omega_{pos} < 50$), confirming that aggressive up-weighting is necessary to handle the sparsity of future concepts. (See Appendix A for full diagnostic tables)

### 6.8. Robustness: Web-Scale Text Evaluation

To address concerns that our model may be overfitted to the clean FineWeb-Edu distribution, we evaluated all models on a 10k-sample subset of C4 (Raffel et al., 2020)—a noisier, web-scale corpus. Results in Table 5 show that while all models degrade on this harder distribution, our Idea-Gated model maintains its relative advantage, suggesting the semantic planning mechanism is robust to domain shift.

*Table 5.* **C4 Robustness Evaluation.** Performance on web-scale text (10k samples). Our model maintains relative advantages despite domain shift, though all models experience degradation on this noisier corpus.

| Model | PPL $\downarrow$ | $S_p \uparrow$ |
|---|---|---|
| QLoRA Control | $12.34_{0.08}$ | 0.59 |
| MTP | $12.21_{0.09}$ | 0.60 |
| **Idea-Gated (Ours)** | $\mathbf{12.08}_{0.11}$ | **0.63** |

## 7. Discussion

The Entropy-Adaptive Idea-Gated Transformer represents a shift from purely associative modeling to constrained semantic planning. Our results suggest that the pervasive limitations of Next-Token Prediction (NTP) are not inherent to the Transformer architecture itself, but rather a byproduct of an objective function that lacks an explicit global planning constraint.

### 7.1. Mechanism of Action: Entropy-Aware Gating

A central design principle of this architecture is **Adaptive Intervention**. Rather than utilizing an explicit computational switch, we induce an emergent gating behavior through the entropy-weighted objective ($\delta = 2.0$). This penalty forces the Idea Head to minimize its footprint during deterministic transitions, reserving its capacity for high-uncertainty semantic shifts.

During inference, this manifests as implicit load-balancing. In regimes of **Syntactic Pass-Through** (e.g., function words), the Idea Head converges to a high-entropy flat distribution which, due to softmax shift-invariance, exerts negligible influence on the backbone. Conversely, during **Semantic Intervention** at branching points, the Head produces a sharp, low-entropy signal; here, the Variance Alignment mechanism magnifies these sparse logit peaks to actively prune the search space and enforce the semantic plan.

### 7.2. Comparison with Multi-Token Prediction

Our results demonstrate that predicting the *semantic core* (bag-of-words) outperforms predicting the *syntactic sequence* (multi-token prediction) for drift reduction. We hypothesize this occurs because:

1. **Lower Variance Signal:** BoW invariance to word order reduces objective variance, providing a stable learning signal.
2. **Semantic Commitment:** Targeting concepts avoids the "syntactic trap" (local coherence but global drift).
3. **Graceful Degradation:** Under high uncertainty, BoW degrades to topic prediction rather than failing.

### 7.3. The Topological Core of "Ideas"

Standard Multi-Token Prediction (MTP) architectures force models to predict the exact *sequence* of future tokens. We argue that this is a suboptimal objective for planning, as it forces the planner to learn syntactic noise (e.g., the precise order of adjectives) alongside semantic intent.

By utilizing a **Bag-of-Words (BoW)** objective for the Idea Head, we capture the "Topological Core" of the future context—the set of concepts that must exist regardless of the specific syntactic path chosen. This relaxes the learning objective: the model predicts *what* will be said without being

penalized for not knowing exactly *when* it will be said.

### 7.4. Variance Alignment as a Control Interface

The success of our approach relies heavily on the **Logit-Space Variance Alignment** introduced in Section 4.3. A fundamental issue in augmenting pre-trained LLMs is the magnitude gap between logits of Idea Head and Token Head.

Our **Asymmetric Integration Protocol** effectively solves this problem. We apply **Mean Centering** consistently across both phases to maintain distributional alignment. However, we strictly decouple the *magnitude* control: we disable variance scaling during training to isolate the learning signal, and activate **Variance Alignment** only during inference. This grants the planner the necessary "authority" to override the backbone's strong unigram priors during deployment, offering a generalizable blueprint for grafting new capabilities onto frozen foundation models.

### 7.5. Efficiency and Democratization

Unlike Chain-of-Thought (CoT) (Kojima et al., 2022; Yao et al., 2023) methods that increase inference costs linearly with reasoning depth (Wei et al., 2022), our approach shifts the planning burden to the latent space. The parallel execution of the Idea Head adds constant-depth computational overhead to the decoding step. While this involves an additional projection over the vocabulary, it avoids the recursive latency of generating intermediate reasoning tokens.

Furthermore, by restricting trainable parameters to the LoRA adapters and the Idea MLP, we enable the development of "System 2" capabilities on consumer-grade hardware (24GB VRAM). This demonstrates that advanced semantic control does not require massive compute clusters, but rather smarter architectural alignment between generation and planning objectives.

### 7.6. Limitations and Future Work

**Scale and Scope:** Our evaluation is limited to the 7B parameter scale. While we demonstrate significant improvement in Semantic Persistence, future work must verify if larger models (which inherently possess stronger semantic priors) require different gating thresholds ($\alpha$). We are currently conducting experiments on Mistral-13B and preliminary results suggest the approach scales favorably.

**Confident Hallucination:** A distinct failure mode arises when the backbone is *confidently wrong* (Low Entropy, High Error). In these cases, the entropy-adaptive gate remains dormant. Preliminary experiments suggest that triggering intervention based on *Semantic Disagreement* (i.e., when $z_{token}$ and $z_{idea}$ are orthogonal) could mitigate this, though this introduces a trade-off between correction and over-suppression.

## 8. Conclusion

We presented the **Entropy-Adaptive Idea-Gated Transformer**, a novel architecture that enforces semantic coherence by decoupling the processes of **semantic macro-planning** and **syntactic execution**. By augmenting a frozen **Mistral-7B** backbone with a **parameter-efficient** auxiliary Idea Head, we created a differentiable semantic bottleneck that dynamically prunes the search space in favor of future intent.

Our work introduces two critical technical innovations to the field of controllable generation. First, our **Entropy-Aware Training** objective instills an **Adaptive Intervention** mechanism, teaching the planner to specialize in high-uncertainty semantic junctions while remaining dormant during deterministic transitions. Second, our **Logit-Space Variance Alignment** strategy solves the fundamental scale mismatch between pre-trained backbones and auxiliary heads, enabling the planner to intervene effectively without destabilizing the generation.

Experiments on **FineWeb-Edu** demonstrate that this framework substantially mitigates *Associative Drift* ($S_p : 0.64 \rightarrow$ **0.71**) while improving perplexity (**7.78**). These gains, alongside improvements on **ARC-Challenge** and **TruthfulQA**, suggest that the limitations of current LLMs are not inherent to the Transformer architecture, but to the flat Next-Token Prediction objective. By forcing models to "think in ideas" via an entropy-aware bottleneck, we offer a robust path toward models grounded in global intent rather than local statistics.

## Impact Statement

This work advances the field of **Interpretable & Controllable Generation** by validating that autoregressive models can be effectively steered via a transparent semantic bottleneck. Unlike opaque activation steering techniques, our Idea Head generates human-readable concept plans prior to token generation. This **Auditability** is a crucial step toward safer AI systems, as it allows observers to detect and intervene on misaligned model intent (e.g., a plan to generate toxic content) before a single word is written.

**Efficiency & Democratization:** Our method promotes **Sustainable AI** by enhancing the capabilities of 7B-class models without the prohibitive cost of full-scale pre-training or scaling to 70B+ parameters. Although the auxiliary head adds parameters, the computational overhead is memory-bandwidth bound rather than compute-bound. Consequently, this architecture remains deployable on consumer-grade hardware (e.g., single 24GB GPU), lowering the barrier to entry for academic research into reasoning-enhanced LLMs and reducing the carbon footprint associated with training large-scale control vectors.

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

## A. Idea Head Diagnostics

We evaluate the task of Bag-of-Words prediction on a hold-out set of 10,000 sequences.

### A.1. Precision, Recall, and F1

Since the target vector is sparse (approx. 20 active concepts per window of size $V$), standard accuracy is a poor metric. We report Precision and Recall for the top-$K$ predicted concepts ($K = 20$) against the ground truth future window.

*Table 6.* Idea Head Predictive Performance (Validation Split).

| Metric | Step 1k (Warmup) | Step 25k | Step 50k (Final) |
|---|---|---|---|
| Precision@20 | 0.12 | 0.38 | **0.45** |
| Recall@20 | 0.15 | 0.41 | **0.52** |
| F1 Score | 0.13 | 0.39 | **0.48** |

The F1 score of 0.48 is significantly higher than a random baseline ($< 0.03$) and indicates that while the model does not perfectly predict every future word (which is impossible due to synonymy), it reliably captures the *semantic core* of the future paragraph.

## B. Human Evaluation Protocol

### B.1. Annotation Guidelines

We recruited 3 expert annotators (graduate students in NLP with 3+ years of experience) and provided detailed guidelines:

**Coherence (1-5):**

- 1 = Incoherent, topic switches abruptly
- 3 = Mostly coherent with minor drift
- 5 = Perfectly coherent, maintains topic throughout

**Factuality (1-5):**

- 1 = Contains clear factual errors
- 3 = Mostly accurate with minor issues
- 5 = Fully factually accurate

**Topic Adherence (1-5):**

- 1 = Drifts to unrelated topic
- 3 = Stays loosely on topic
- 5 = Perfectly adheres to intended topic

### B.2. Inter-Annotator Agreement

We computed Cohen's $\kappa$ across all three dimensions:

- Coherence: $\kappa = 0.61$ (substantial agreement)
- Factuality: $\kappa = 0.68$ (substantial agreement)
- Topic Adherence: $\kappa = 0.63$ (substantial agreement)

These high agreement scores indicate that the improvements are genuine and not artifacts of annotator variance.

## C. Inference Latency Analysis

We measured inference latency on a single NVIDIA A100 (80GB) with batch size=1 using a standard PyTorch implementation (Hugging Face).

Our method avoids the prohibitive $2\times$ compute cost of Contrastive Decoding. The $13\%$ latency increase is primarily due to the memory-bandwidth cost of loading the Idea Head parameters ($\sim$296MB in fp16) during the decoding step.

*Table 7.* Inference Latency vs. Baselines. Unlike Contrastive Decoding, which requires two forward passes, our method incurs only a minor overhead for the auxiliary head projection.

| Model | Throughput (tok/s) | Relative Slowdown |
|---|---|---|
| Mistral-7B (Base) | 42.5 | - |
| Contrastive Decoding | 22.8 | 46.4% (approx. 2x cost) |
| **Idea-Gated (Ours)** | **36.8** | **13.4%** |

# D. Additional Ablation Studies

## D.1. Gating Coefficient Sweep

*Table 8.* Impact of gating coefficient $\alpha$ on model performance. Standard deviations are computed over 3 random seeds. We observe that $\alpha = 0.5$ provides the optimal balance, maximizing Semantic Persistence ($S_p$) while minimizing Perplexity (PPL). Extreme values ($\alpha = 1.0$) degrade performance by over-constraining the backbone.

| $\alpha$ | **PPL** $\downarrow$ | $S_p \uparrow$ |
|---|---|---|
| 0.0 (No gating) | $8.07_{0.03}$ | 0.64 |
| 0.3 | $7.91_{0.05}$ | 0.68 |
| **0.5 (Ours)** | $\mathbf{7.78_{0.04}}$ | **0.71** |
| 0.7 | $7.82_{0.06}$ | 0.70 |
| 1.0 | $7.97_{0.07}$ | 0.67 |

At $\alpha = 1.0$, the semantic gate becomes too aggressive, occasionally rejecting grammatically necessary function words, leading to degraded fluency (higher PPL). At $\alpha = 0.5$, we achieve optimal balance.

## D.2. Positive Class Weight Sensitivity

*Table 9.* Impact of positive class weight $\omega_{pos}$ on Idea Head learning. Performance is stable across the [100, 300] range, but degrades at extreme values where the loss either ignores rare concepts ($\omega = 50$) or overfits to noise ($\omega = 500$).

| $\omega_{pos}$ | **F1@20** | $S_p \uparrow$ |
|---|---|---|
| 50 | $0.33_{0.02}$ | 0.66 |
| 100 | $0.44_{0.03}$ | 0.69 |
| **200 (Ours)** | $\mathbf{0.48_{0.02}}$ | **0.71** |
| 300 | $0.46_{0.03}$ | 0.70 |
| 500 | $0.37_{0.04}$ | 0.68 |

Performance is stable across $\omega_{pos} \in [100, 300]$, but degrades at extreme values where the head either under-commits ($\omega_{pos} < 100$) or over-commits ($\omega_{pos} > 300$) to concept predictions.

# E. Failure Case Analysis

**Extremely Long-Range Dependencies** Our $K = 20$ window captures medium-range context but may miss very long-range dependencies (e.g., chapter-level narrative consistency in books). Adaptive window sizing based on detected topic boundaries could address this.

# F. Computational Requirements

## F.1. Training

- **Hardware:** Single NVIDIA RTX 4090 (24GB VRAM)
- **Training Time:** $\sim$72 hours for 50k iterations
- **Memory Usage:** Peak 22.3GB VRAM (with gradient accumulation)
- **Total Cost:** $\sim$\$50 USD on cloud GPU (RunPod)

## F.2. Inference

- **Additional Memory:** +148M parameters (296MB in fp16 and 592MB in fp32)
- **Latency Overhead:** 13% slowdown vs base model

- **Throughput:** 36.8 tok/sec on A100 (batch size=1)

The modest computational requirements make this approach accessible to academic researchers without institutional compute clusters.

## G. Broader Impacts and Ethical Considerations

### G.1. Positive Impacts

- **Reduced Misinformation:** By maintaining topical coherence, the model is less likely to drift into false or misleading tangents.
- **Interpretability:** The Idea Head's BoW predictions are human-readable, enabling inspection of the model's planning process.
- **Accessibility:** Low computational requirements democratize access to controllable generation research.

### G.2. Potential Concerns

- **Over-Steering Risk:** If miscalibrated, the gate could suppress valid creative tangents, leading to overly rigid generation.
- **Dual Use:** While improved coherence generally benefits users, it could also make generated misinformation more convincing if the base model has incorrect beliefs.
- **Training Data Bias:** FineWeb-Edu's educational focus may introduce domain-specific biases. Evaluation on diverse corpora is critical.

### G.3. Mitigation Strategies

We recommend:

1. Combining this approach with factuality verification systems (e.g., retrieval-augmented generation)
2. Providing users with transparency into when the gate intervenes
3. Continued evaluation on diverse datasets and demographic groups

