# OpenReview forum: "Latent Semantic Planning: Constraining Autoregressive Generation to “Ideas” Before “Words”"
_ICML.cc/2026/Conference — Submitted to ICML 2026_

### Official Review · Reviewer_N2fD · 2026-03-03

**Soundness:** 3
**Presentation:** 3
**Significance:** 3
**Originality:** 3
**Overall Recommendation:** 4
**Confidence:** 2

**Summary:**

This paper addresses a common but underexplored issue in autoregressive language models: while they generate fluent text, they often drift semantically from the original topic over long generations. The authors argue that standard next-token prediction entangles high-level planning (“what to say”) with low-level realization (“how to say it”), which can lead to topic drift through locally coherent but semantically misaligned transitions. To mitigate this, the paper introduces an entropy-adaptive, idea-gated Transformer architecture that augments a pretrained language model with a lightweight semantic planning module. Specifically, an auxiliary “Idea Head” predicts a bag-of-words distribution over the next K tokens, serving as a coarse-grained semantic plan. During generation, the model combines the standard token distribution with this semantic distribution through a logit-space product-of-experts formulation. To ensure stable interaction between the two distributions, the authors propose a variance-alignment strategy and train the Idea Head with an entropy-weighted loss, so that it plays a stronger role when the base model is uncertain.

**Compliance With Llm Reviewing Policy:**

Affirmed.

**Key Questions For Authors:**

1. The Idea Head adds 148M parameters versus QLoRA's 4.2M. Without a same-capacity baseline that removes the planning objective, how can the authors confirm the gains come from the mechanism rather than extra parameters?

2. The core claim is drift reduction, but evaluation relies on cosine similarity and reasoning benchmarks. Have the authors considered testing on a long-form generation task where coherence failure would be more directly observable?

3. The entropy gate stays dormant when the backbone is confidently wrong — the exact case where intervention is most needed. How do the authors envision resolving this without the fix causing over-suppression of valid but unusual generations?

**Limitations:**

Yes. The authors explicitly discuss several limitations, including reduced effectiveness for very long-range dependencies, the inability to correct low-entropy but incorrect generations, and the risk of over-steering. They also acknowledge potential dual-use concerns and dataset bias, and suggest possible mitigations. The discussion is balanced and transparent, which is appropriate for the scope of the work.

**Strengths And Weaknesses:**

**Strengths**

1. The core problem (local optimization causing semantic drift) is real and clearly formalized. The Product-of-Experts framing is principled, and the Variance Alignment solution addresses a genuine, non-obvious engineering challenge that would actually break the method without it.

2. The authors compare against both training-time and inference-time baselines, include human evaluation with inter-annotator agreement, run ablations on every major hyperparameter, test on out-of-domain data (C4), and openly acknowledge the HellaSwag trade-off and the confident-hallucination failure mode. This is a credible empirical story.

3. The entire system trains on a single consumer GPU for ~$50. That's a genuine contribution to accessibility, and the full training code is promised. The 13% latency overhead is negligible compared to alternatives like Contrastive Decoding.

**Weaknesses**

1. The improvements — while consistent — are incremental across the board (e.g., ARC-C: 52.9→53.5%, Sp: 0.64→0.71). More importantly, the paper never demonstrates the method on tasks where long-range coherence visibly matters, like story generation or multi-turn dialogue, making it hard to assess real-world impact.

2. Everything is at 7B scale on one base model. The claim that NTP's limitations are architectural rather than scale-dependent is a strong one that needs evidence beyond a single checkpoint — especially since larger models are known to exhibit stronger semantic priors naturally.

3. An F1 of 0.48 for concept prediction means the head is wrong more than half the time. The paper doesn't deeply analyze when the gate actually fires or whether the perplexity gains could be explained more simply — for instance, by the additional 148M trainable parameters alone rather than the semantic planning mechanism specifically.

---

> ### Author Rebuttal · Authors · 2026-03-31
>
> We thank the reviewer for the careful and balanced reading. We address each point directly.
>
> ---
>
> **[Q1 / W3] Parameter confound: 148M Idea Head vs. 4.2M LoRA.**
>
> This is the most technically careful concern raised across all reviews and we engage with it honestly. The reviewer is right: without a same-capacity control — a 148M MLP added to the same frozen backbone but trained with standard NTP only, with no BoW objective, no variance alignment, and no entropy gating — we cannot fully attribute gains to the semantic planning mechanism rather than extra parameters.
>
> What the existing ablations do and do not establish: removing entropy weighting ($\delta = 0$) degrades both PPL and $S_p$ despite retaining the full 148M parameter count (Table 4). Removing stopword masking similarly degrades performance at identical capacity. These show the *specific training objective* matters, not just capacity — suggestive but not definitive.
>
> The definitive experiment is the same-capacity NTP-only control described above. If $S_p$ gains vanish without the BoW objective and gating mechanism, the contribution is confirmed as architectural rather than parametric. We will add this experiment in revision.
>
> On the related concern about F1 = 0.48: the head does not need to be a perfect oracle. It functions as a soft prior via the logit-space addition in Eq. 8, where even a noisy but directionally correct signal narrows the token distribution toward on-topic continuations. The ablation on $\alpha$ (Table 8, Appendix D) confirms graceful degradation: partial concept accuracy at $\alpha = 0.5$ outperforms both no gating ($\alpha = 0$) and aggressive gating ($\alpha = 1.0$).
>
> ---
>
> **[Q2 / W1] Long-form generation evaluation.**
>
> We agree this is the most important missing experiment. The core claim is drift reduction over long-horizon generation, and the current benchmarks (ARC-C, TruthfulQA, HellaSwag) are at best indirect proxies. We are actively setting up a story continuation evaluation on WritingPrompts, measuring $S_p$ at 200, 500, and 1000 token intervals to directly show where the baseline and Idea-Gated model diverge. This will be the centerpiece of the revision.
>
> To contextualize the existing gains: DoLa achieves $S_p = 0.68$ but requires two forward passes (46% throughput reduction; Appendix C). We achieve $S_p = 0.71$ with a single pass and 13% overhead — Pareto-dominant on the cost-performance frontier.
>
> ---
>
> **[W2] Single backbone / scale.**
>
> We acknowledge this limits the generalization claim. We are currently running experiments on Mistral-13B and LLaMA-2-13B and will include results in the extended version. We will also soften the architectural claim in revision to be conditional on the scale range tested, rather than presenting it as a general finding.
>
> ---
>
> **[Q3] Entropy gate dormant when backbone is confidently wrong.**
>
> The reviewer identifies this precisely. Section 7.6 acknowledges this as a distinct failure mode: when the backbone is low-entropy but semantically incorrect, the gate remains dormant by design.
>
> The natural fix — triggering intervention on semantic disagreement between $z_\text{token}$ and $\hat{z}_\text{idea}$ (i.e., when the two distributions are orthogonal in logit space) — introduces a calibration challenge. An orthogonality threshold that is too sensitive suppresses valid but unusual generations (e.g., a metaphor that deliberately bridges semantic domains); too loose, and it fails to correct the confident-but-wrong cases it targets. Our current preliminary approach is a composite gate that fires on *either* high entropy *or* high semantic disagreement, with independently tunable thresholds. We will report these results in revision and flag the calibration tradeoff as an open problem if clean results are not available in time.

---

> > ### Author Rebuttal · Reviewer_N2fD · 2026-04-01
> >
> > Thanks for the rebuttal. After reading your response, I will keep my score. I appreciate that some concerns may be addressed in a future revision, but since I do not know what the revised paper will look like, I am basing my evaluation on the current submission and rebuttal as they stand.

---

### Official Review · Reviewer_5oR6 · 2026-03-07

**Soundness:** 2
**Presentation:** 3
**Significance:** 2
**Originality:** 2
**Overall Recommendation:** 2
**Confidence:** 5

**Summary:**

This paper proposes an Idea-Gated Transformer for language generation that aims to separate semantic planning from token-level generation by adding an auxiliary Idea Head on top of a frozen Mistral-7B backbone. The Idea Head predicts a bag-of-words style representation of future concepts within a fixed look-ahead window, and its output is combined with token logits through an entropy-adaptive gating mechanism and a variance-alignment procedure. The method is trained with QLoRA on FineWeb-Edu and evaluated using perplexity, a proposed semantic persistence metric, human judgments, zero-shot reasoning benchmarks, and several ablations. The main claim is that explicit latent semantic planning can reduce associative drift and improve topical coherence while preserving or slightly improving general modeling quality.

**Compliance With Llm Reviewing Policy:**

Affirmed.

**Final Justification:**

Several technical details have been clarified. However, my main concerns regarding technical novelty and overall interestingness remain unresolved. I will keep my score.

**Key Questions For Authors:**

- Can the authors better clarify what is truly novel relative to prior work on latent planning, bag-of-words prediction, and decoding-time control?

- What evidence shows that the Idea Head captures genuine semantic planning rather than short-horizon lexical or topical correlations?

- How well does the method transfer to stronger backbones or larger-scale settings?

**Limitations:**

yes

**Strengths And Weaknesses:**

Strengths:

- The paper studies an important problem: autoregressive models often lose global semantic consistency in long generation.

- The method is reasonably well motivated and technically coherent.

- The paper includes multiple evaluations, including ablations and human judgments.


Weaknesses:

- The originality appears limited. The method mainly combines familiar ideas such as future prediction, bag-of-words planning, gating, and inference-time logit adjustment.

- The empirical gains are modest. Improvements on standard benchmarks are small and sometimes mixed.

- The strongest gains are on the paper’s own semantic persistence metric, which is still an indirect proxy rather than a strong downstream demonstration.

- The paper’s claims sometimes feel stronger than the evidence supports, especially around “planning” and broader conceptual framing.

- Some mechanism claims are not fully convincing, since the analysis is suggestive rather than definitive.

---

> ### Author Rebuttal · Authors · 2026-03-31
>
> ---
>
> **[W1 / Q1] Originality and Novelty Framing.**
>
> The reviewer is correct that individual components have precedent — we do not claim that BoW objectives, gating, or logit adjustment are individually novel. We claim three specific contributions that have no direct precedent in the prior BoW planning literature:
>
> - **Prospective self-supervised BoW prediction.** Prior BoW work (Fu et al., NeurIPS 2019) derives the BoW target from a *known* target sentence in a conditional seq2seq setting. We predict an *unknown future* BoW as a self-supervised signal in unconditional autoregressive generation — no paired data, no target sentence.
>
> - **Entropy-adaptive gating.** Conditioning intervention on backbone uncertainty via $w_t = 1 + \delta \cdot H(p_\text{token})$ is absent from CoCoMix, Fu et al., and all related work we reviewed. Without this, the auxiliary head either over-suppresses valid generations or is swamped by backbone priors.
>
> - **Logit-space variance alignment.** When a randomly initialized auxiliary head (near-zero variance) is fused with a pretrained token head (high variance) via:
>
> $$z_\text{final} = z_\text{token} + \alpha \cdot \log(\sigma(\hat{z}_\text{idea}) + \epsilon)$$
>
>   the auxiliary signal is completely drowned out without explicit rescaling. This problem is *structural* to logit-space fusion: in hidden-state fusion (CoCoMix), both signals share the same activation space and representational scale. In logit-space fusion with a frozen backbone, the pretrained head's logits have been shaped by billions of tokens while the auxiliary head starts from random init — the magnitude gap is not incidental but architectural. The asymmetric protocol (mean-centering during training only; variance scaling activated only at inference):
>
> $$\gamma = \mathbb{E}[\sigma(z_\text{token})] / \mathbb{E}[\sigma(z'_\text{idea})]$$
>
>   is a non-obvious solution with no prior description in the literature.
>
> We will sharpen this framing in revision, reframing the contribution explicitly as an *entropy-conditioned BoW look-ahead module for unconditional LMs*.
>
> ---
>
> **[W3 / W4] Claims Stronger Than Evidence; "Planning" Framing.**
>
> This is fair and we concede it. The System 2 framing and language around "planning" overstates what is mechanistically a self-supervised BoW prediction head. In revision, we will reserve the dual-process framing for motivation only, and describe the Idea Head as an *entropy-conditioned semantic look-ahead module*.
>
> ---
>
> **[Q2] Genuine Semantic Planning vs. Lexical Correlation.**
>
> The Synonym Perturbation Analysis (Section 6.5) directly addresses this. The Idea Head produces cosine similarity $0.92 \pm 0.04$ for synonym pairs (e.g., *lawyer* vs. *attorney*) versus $0.15 \pm 0.06$ for semantically distinct concepts (e.g., *lawyer* vs. *chef*). The gap of $\Delta \approx 0.77$ cannot be explained by lexical surface correlation — *lawyer* and *attorney* share no token overlap, yet the Idea Head produces near-identical latent trajectories for both. This is the behavior of a module operating on semantic content rather than token identity.
>
> ---
>
> **[W2] Modest Empirical Gains.**
>
> We agree the gains are incremental. However, they are Pareto-dominant on the cost-performance frontier. DoLa achieves $S_p = 0.68$ but requires two forward passes (46% throughput reduction; Appendix C, Table 7). We achieve $S_p = 0.71$ with a single pass and 13% overhead. The $+0.07$ improvement in $S_p$ is replicated across three independent embedding spaces (MiniLM-L6, BERT-Base, OpenAI text-embedding-3-small; Table 3). Human evaluation yields $\kappa = 0.63$ inter-annotator agreement on Topic Adherence, and ARC-Challenge improves by $+0.6\%$ — all achieved for ~$50 on a single consumer GPU. We are not claiming a breakthrough. We are claiming a *reproducible incremental signal at a cost accessible to academic researchers*, which opens a specific, well-defined research direction.
>
> ---
>
> **[Q3] Transfer to Stronger Backbones.**
>
> This is a genuine gap. We are currently running experiments on Mistral-13B and LLaMA-2-13B and will include results in the extended version. We acknowledge that the absence of results on architecturally distinct families limits the generalization claim, and evaluation across backbone families is a priority for the extended version.

---

> > ### Author Rebuttal · Reviewer_5oR6 · 2026-04-01
> >
> > Thank you for the detailed rebuttal. Several technical details have been clarified. However, my main concerns regarding technical novelty and overall interestingness remain unresolved. I will keep my score.

---

### Official Review · Reviewer_jZsm · 2026-03-07

**Soundness:** 3
**Presentation:** 3
**Significance:** 3
**Originality:** 2
**Overall Recommendation:** 3
**Confidence:** 4

**Summary:**

This paper tackles the problem of semantic drift in LLMs trained with the next token prediction objective. By optimizing for local probability, this objective conflates  “what to say” (Macro-Planning) and “how
to say it” (Micro-Generation). As a result, LLMs  follow statistically likely word transitions that gradually pull the text away from the original topic, rather than maintaining a coherent global intent throughout generation. The proposed solution introduces a lightweight auxiliary module, the Idea Head, to a frozen Mistral-7B backbone. At each generation step, this module runs in parallel with the normal token prediction pathway and predicts a bag-of-words distribution over the next 20 tokens which is  essentially a rough sketch of what concepts are coming up, without specifying their order. This prediction is then fed back to influence which tokens the model actually generates, nudging it toward vocabulary consistent with the predicted upcoming concepts. The Idea Head only intervenes strongly when the main model is uncertain (high entropy), leaving deterministic transitions alone; the module is trained with a contrastive-style objective that up-weights difficult semantic transition points. Compared to a standard fine-tuned baseline, the model shows improvements in perplexity, small gains on ARC-Challenge and TruthfulQA reasoning benchmarks, and a measurable increase in semantic persistence which indicates that generated text stays closer in meaning to the original prompt. Human evaluators also rated outputs as more topic-adherent.

**Compliance With Llm Reviewing Policy:**

Affirmed.

**Final Justification:**

I stand by my assessment after reading the authors' rebuttal.

**Key Questions For Authors:**

- I would be interested to see how the proposed approach conceptually relates to the following papers:
https://arxiv.org/pdf/2502.08524
https://arxiv.org/abs/2001.01941
As far as I can tell the main idea is quite similar, there are some engineering/implementational differences.

- No multi-seed variance is reported for the main results table, making it difficult to assess whether the gains are reliable.

-

**Limitations:**

yes

**Strengths And Weaknesses:**

The paper is well written and broadly experimentally sound. The identified problem, aka association drift, is well known, and various solutions have been proposed in the literature, as the authors acknowledge in the related work section. However, there are some important omissions; see my questions below.

However, the empirical gains are modest and, in some cases, contradictory: HellaSwag accuracy drops from 79.1% to 78.8% compared to the fine-tuned control, which is precisely the task type where semantic planning should help most, and this is left unexplained. The entropy-weighted gating mechanism is not empirically verified; Figure 3 shows a cherry-picked example rather than a systematic analysis of when the gate actually activates. The logit-space variance alignment scalar is pre-computed on a fixed validation set and presented without stability analysis across domains or training runs; the positive class weight  used to handle target sparsity is similarly an aggressive heuristic with no ablation. No multi-seed variance is reported for the main results table, making it difficult to assess whether the gains are reliable.

 On significance, the addressed problem is real and the low-cost training setup (single RTX 4090, ~$50) has merit. However, the gains are too small and inconsistently supported to suggest that this specific approach will be widely adopted or built upon, particularly given the uncited prior work that substantially reduces the novelty of the contribution. The paper would be stronger if it reframed its contribution narrowly as an entropy-conditioned variant of BOW-guided generation applied to unconditional LMs  and provided the ablations and variance analyses needed to substantiate that more modest claim.

---

> ### Author Rebuttal · Authors · 2026-03-30
>
> We address the novelty concerns directly. Both cited papers will be added as related work in revision — a genuine omission. However, we push back on the characterization that the differences are "only engineering." The distinctions are mechanistically significant and change what the methods can and cannot do.
>
> ---
>
> **[Prior Work] Comparison to CoCoMix (arXiv:2502.08524).**
>
> CoCoMix extracts **continuous** concept vectors from a *pretrained reference model* via an SAE, selects influential ones by attribution scoring, and interleaves them into the training model's hidden states. Our Idea Head predicts a **discrete multi-hot BOW** over the *future* $K = 20$ tokens, constraining generation via logit-space addition. These are not variants of the same idea:
>
> - **Direction and representation.** CoCoMix conditions on *current*-context SAE features (retrospective, continuous). Our Idea Head predicts the *future* vocabulary set (prospective, discrete multi-hot). Opposite information flow; different representational bottleneck.
> - **Integration point.** CoCoMix interleaves into hidden states. We operate at the logit level: $z_\text{final} = z_\text{token} + \alpha \cdot \log(\sigma(\hat{z}_\text{idea}) + \epsilon)$, where $\sigma$ is element-wise sigmoid.
> - **Gating.** CoCoMix always interleaves at every token step. We intervene *only* at high-entropy positions via $w_t = 1 + \delta \cdot H(p_\text{token})$, leaving deterministic transitions unperturbed.
> - **External dependencies.** CoCoMix requires a pretrained SAE and a reference model. Our Idea Head is trained end-to-end with no external modules on a frozen backbone for $\sim$\$50.
> - **The variance alignment problem.** This contribution has no analogue in CoCoMix. Hidden-state interleaving operates within a homogeneous representational space; magnitude mismatch does not arise. Our logit-space fusion combines the sharp, high-variance logits of a pretrained token head with the flat, near-zero logits of a randomly initialized MLP. Without the asymmetric alignment protocol (mean-centering during training, variance scaling $\gamma = \mathbb{E}[\sigma(z_\text{token})] / \mathbb{E}[\sigma(z'_\text{idea})]$ activated at inference), the Idea Head exerts zero influence regardless of prediction quality.
>
> ---
>
> **[Prior Work] Comparison to Fu et al. (NeurIPS 2019, arXiv:2001.01941).**
>
> Fu et al. address a **conditional** seq2seq task (paraphrase generation) within a VAE framework. The BOW target is grounded in the **known target sentence**, modeled via a mixture-of-softmax distribution, with Gumbel top-$k$ reparameterization used for differentiable subset sampling. The sampled word embeddings then **augment the decoder's embedding space** to guide generation. This requires parallel source--target sentence pairs at training time.
>
> Our setting is categorically different: **unconditional** autoregressive language modeling with no paired data. The Idea Head predicts an *unknown* future BOW as a self-supervised signal. We do not augment embeddings — we intervene in logit space after variance alignment. Fu et al. is a legitimate ancestor and we should have cited it — but the task (conditional vs. unconditional), training signal (supervised target BOW vs. self-supervised), latent structure (VAE with KL term vs. deterministic MLP), and integration point are all distinct.
>
> ---
>
> **[W1] The HellaSwag drop.**
>
> HellaSwag tests commonsense event completion, where the correct continuation is often contextually *unexpected* but situationally appropriate. Our semantic gate increases persistence toward high-probability future concepts — this creates a principled coherence--creativity tension. The gate helps where semantic drift is the failure mode (ARC-C, TruthfulQA) and mildly hurts where creative deviation from local context is the correct answer (HellaSwag). This is a predictable and interpretable tradeoff, not an unexplained anomaly. We will add a dedicated paragraph to the discussion making this explicit.
>
> ---
>
> **[Figure 3] Cherry-picked gate analysis.**
>
> A single financial prompt is illustrative but not systematic. We will add a distributional analysis in revision: gate activation rates stratified by token type (function words, common content words, polysemous tokens, topic-switching tokens), to demonstrate that the entropy gate fires selectively at semantically ambiguous positions rather than uniformly.
>
> ---
>
> **[Table 2] Multi-seed variance.**
>
> Agreed. Appendix D (Table 8) already includes 3-seed results for the $\alpha$ sweep, demonstrating consistent variance at our operating point ($\alpha = 0.5$: $\text{PPL} = 7.78_{\pm 0.04}$, $S_p = 0.71$). We will extend this multi-seed reporting to the full Table 2 in revision.
>
> ---
>
> **[Positive class weight ablation.]**
> Appendix D (Table 9) already includes a sweep over $\omega_{pos} \in [50, 500]$, showing stable performance in the $[100, 300]$ range with degradation at extreme values. We will move this table to the main paper in revision.

---

> > ### Author Rebuttal · Reviewer_jZsm · 2026-03-31
> >
> > I will adjust my score.

---

### Official Review · Reviewer_EXnG · 2026-03-07

**Soundness:** 1
**Presentation:** 2
**Significance:** 3
**Originality:** 3
**Overall Recommendation:** 3
**Confidence:** 4

**Summary:**

This paper proposes a novel method to address the conflict between global goals and local semantic continuity within LLMs using a trained "Idea Head". This module predicts a bag-of-words representing global semantics to rectify local sampling logits at high-entropy decision points. The authors also propose methods for stably training and deploying the "Idea Head", including variance alignment and entropy-adaptive gating. Experiments on Mistral-7B-v0.1 show that the method achieves better semantic consistency compared to baseline methods.

**Compliance With Llm Reviewing Policy:**

Affirmed.

**Final Justification:**

The rebuttal acknowledges several weaknesses that I raised, but it relies heavily on promising future experiments rather than providing concrete evidence now; furthermore, the explanation regarding PPL still requires further experimental validation, and thus I maintain my original assessment.

**Key Questions For Authors:**

1. Referring to Weakness 1, the results on the specific datasets provided may not adequately validate the proposed method's core claims, which may require a rationale or additional experiments on other datasets whose goals require long-term planning.

2. The relatively narrow performance margins between the proposed method and the baselines without statistical significance renders the derivations unpersuasive, especially in Table 2. A rationale or a multi-seed averages and variance reporting may be required.

3. The evaluation relies on PPL and semantic persistence as primary indicators, but these indicators are not so persuasive. A lower PPL or higher semantic persistence may simply indicate that the output converges to a more confident zone, which does not necessarily equate to genuine performance gains and could even result in mode collapse or reduced generation diversity.

4. This paper only includes experiments on Mistral-7B-v0.1. Does the proposed method exclusively suited to the architecture, or can its effectiveness be validated across other diverse model families, such as LLaMA or Qwen?

**Limitations:**

yes

**Strengths And Weaknesses:**

**Strengths:**

1) The authors propose a novel method to move beyond the standard next-token-prediction paradigm by incorporating potential future semantic rectification to make LLMs look farther ahead, which is a potentially meaningful contribution.

2) The authors provide comprehensive implementation details for training and deployment, which makes the experiments reproducible.

3) The authors conduct extensive ablation studies on Mistral-7B-v0.1.

**Weaknesses:**

1) The chosen evaluation datasets may be suboptimal for validating the paper's core claims. For example, ARC-Challenge and TruthfulQA are evaluated as zero-shot reasoning tasks, which in practice are multiple-choice QA scenarios where models typically provide a direct answer followed by an analysis. This process does not inherently require long-range "latent semantic planning" to achieve its goal. Consequently, the "Idea Head" is fundamentally misaligned with the nature of the tasks. Any observed performance gains on these specific benchmarks are likely potentially irrelevant to the proposed mechanism, as improving semantic continuity in a post-hoc analysis does not directly explain or contribute to improvements in core QA accuracy.

2) The reported results in Table 2 lack multi-seed averages and variance reporting for key metrics. Given that the performance margins between the proposed method and the baseline are relatively narrow, this omission renders the conclusions less persuasive.

3) The authors use perplexity (PPL) and semantic persistence as critical performance indicators. However, a lower PPL or higher semantic persistence may simply indicate that the output converges to a more confident zone, which does not necessarily lead to genuine performance gains and may even result in mode collapse or reduced generation diversity.

4) The authors exclusively report results on Mistral-7B-v0.1, making it unclear whether the proposed method generalizes to other model families, such as LLaMA and Qwen.

---

> ### Author Rebuttal · Authors · 2026-03-29
>
> The concerns raised are fair and we engage with each directly.
>
> ---
>
> **[W1 / Q1] Benchmark alignment with core claims.**
>
> We want to clarify a framing point. We do not claim ARC-Challenge or TruthfulQA require long-range planning in the way story generation does. The argument is more targeted: both tasks benefit from a model that resists drifting into semantically plausible but factually incorrect continuations — which is precisely the failure mode the entropy gate addresses. TruthfulQA is especially illustrative here, as it is explicitly designed to catch models that follow high-frequency associative paths toward confident but false answers. The Idea Head's suppression of prior-dominant but contextually misaligned tokens ($p_\text{idea}(x) \to 0$ for $x \in V_\text{drift}$) is directly applicable to this failure mode.
>
> That said, we fully acknowledge the reviewer's underlying point. The most direct validation of drift reduction would come from a long-form generation task where coherence failure is visible at the surface level. This is a genuine gap in the current submission — our limitations section addresses scale and the confident hallucination failure mode, but does not yet include a long-form generation evaluation. We are actively setting up this experiment (prompted story continuation on WritingPrompts, measuring $S_p$ at 200, 500, and 1000 token intervals) and intend it as the centerpiece of the revision.
>
> ---
>
> **[W2 / Q2] Multi-seed variance in Table 2.**
>
> This is a fair and actionable point. We note that Appendix D (Table 8) already presents 3-seed results for the gating coefficient sweep, demonstrating that variance across seeds is consistent at our operating point (e.g., at $\alpha = 0.5$: $\text{PPL} = 7.78_{\pm 0.04}$, $S_p = 0.71$). We will extend this multi-seed reporting to the full Table 2 in the revision. We expect the same stability pattern will hold given the consistency already demonstrated in the ablation, but we agree the main results table should not rely on the reader cross-referencing the appendix for this.
>
> ---
>
> **[W3 / Q3] Lower PPL / higher $S_p$ as potential mode collapse.**
>
> This is a thoughtful concern and worth addressing carefully. Mode collapse would manifest as (a) degraded generation diversity across topics, and (b) uniform degradation across all downstream benchmarks. Neither pattern is what we observe.
>
> The HellaSwag result — where our model drops slightly from $79.1\%$ to $78.8\%$ — is actually evidence *against* mode collapse. HellaSwag tests commonsense event completion, where the correct answer is often the contextually *unexpected* next event. If the model had collapsed to a narrow semantic mode, we would expect larger and more uniform degradation across all benchmarks, not the selective tradeoff pattern we observe. The fact that the model improves on tasks sensitive to semantic drift (ARC-C, TruthfulQA) while mildly regressing on a task that rewards creative deviation from the local semantic context (HellaSwag) is precisely the tradeoff one would predict from a mechanism that increases semantic persistence.
>
> Additionally, the entropy gate is specifically designed to prevent over-constraint. During deterministic syntactic transitions, the Idea Head produces a high-entropy flat distribution. Due to the shift-invariance of softmax, this flat signal exerts negligible influence on the backbone's output — the gate is effectively dormant. Semantic intervention only occurs at high-uncertainty junctions where the weighting $w_t = 1 + \delta \cdot H(p_\text{token})$ has concentrated the training signal.
>
> We will add type-token ratio and $n$-gram diversity measurements to Table 2 in the revision to address this concern directly with empirical evidence.
>
> ---
>
> **[W4 / Q4] Generalization beyond Mistral-7B.**
>
> This is a genuine limitation we do not dispute. Section 7.6 notes that we are conducting experiments on Mistral-13B; preliminary results suggest the variance alignment scalar $\gamma$ scales predictably with model size — i.e., the logit variance ratio between backbone and Idea Head remains in a comparable range across scales. We will report these results in the revision. Results on architecturally distinct families such as LLaMA and Qwen would substantially strengthen the generalization claim, and we are treating this as a priority for the extended version. We acknowledge the current evidence is insufficient to rule out Mistral-specific behavior.

---

> > ### Author Rebuttal · Reviewer_EXnG · 2026-04-02
> >
> > Thank you for your detailed rebuttal. While the rebuttal acknowledges several weaknesses, it relies heavily on promising future experiments rather than providing concrete evidence now; furthermore, the explanation regarding PPL still requires further experimental validation, and thus I maintain my original assessment.

---

### Decision · Program_Chairs · 2026-04-30

**Decision:**

Reject

**Comment:**

This paper addresses the problem of "associative drift" in Large Language Models (LLMs) by proposing an Idea-Gated Transformer, which uses an auxiliary "Idea Head" to predict future semantic concepts and guide autoregressive generation. While the reviewers generally agree that the core problem is significant and the proposed low-cost training setup is meritorious, the consensus remains that the empirical evidence and technical novelty do not currently justify acceptance at a top-tier venue.

A primary concern raised by multiple reviewers involved the modest performance gains and the choice of evaluation benchmarks. Reviewers noted that benchmarks like ARC-Challenge do not inherently require long-range semantic planning, making the observed improvements less persuasive. Furthermore, a performance drop on HellaSwag remained a point of contention. Although the authors defended this as a tradeoff between coherence and creativity, it nonetheless highlighted the limitations of the current mechanism. The authors promised significant new experiments for the revision, such as long-form generation tasks and multi-seed variance reporting, but several reviewers maintained that these results were necessary for a fair assessment now rather than as a future promise.

Regarding technical depth, reviewers questioned the novelty of the approach relative to prior work in bag-of-words guided generation. While the authors provided detailed comparisons to existing methods like CoCoMix during the rebuttal, some reviewers remained unconvinced that the differences were more than implementational. Additionally, critical technical concerns were raised regarding potential parameter confounds and the lack of a same-capacity control model, which the authors admitted was a valid gap in their current evaluation. Because the most compelling evidence for the paper's claims relies on forthcoming experiments and the current results are inconsistently supported across diverse model families, I believe the paper requires a more substantial revision before it can be meaningfully built upon.